# Online Transfer and Adaptation of Tactile Skill: A Teleoperation Framework

**Xiao Chen,**\*   **Tianle Ni,**\*   **Kübra Karacan,**   **Hamid Sadeghian,**   **Sami Haddadin**

Munich Institute of Robotics and Machine Intelligence, Technical University of Munich

`xiaoyu.chen@tum.de, tianle.ni@tum.de`

**Abstract:** This paper presents a teleoperation framework designed for online learning and adaptation of tactile skills, which provides an intuitive interface without the need for physical access to an execution robot. The proposed tele-teaching approach utilizes periodical Dynamical Movement Primitives (DMP) and Recursive Least Square (RLS) for generating tactile skills. An autonomy allocation strategy, guided by learning confidence and operator intention, ensures a smooth transition from human demonstration to autonomous robot operation. Our experimental results with two 7 Degree of Freedom (DoF) Franka Panda robots demonstrate that the tele-teaching framework facilitates online motion and force learning and adaptation within a few iterations.

**Keywords:** Learning from Demonstration, Online Adaptation, Tactile Skill, teleoperation, Autonomy Allocation

## 1 Introduction

As robots become increasingly integrated into daily activities such as supporting household chores and elderly care [1], the complexity of their tasks continues to grow. Learning from Demonstration (LfD) [2] has emerged as an efficient method to directly transfer expert skills to the robot, thereby reducing the time required for self-learning [3]. Several approaches, including first-order Dynamical Systems [4], Gaussian Mixture Model [5], and Dynamical Movement Primitives (DMP) [6, 7, 8] have been employed to encode the motion skills. However, motion skills alone are insufficient for complex tasks involving rich contact with the environment. Therefore, the incorporation of force skills is also essential. In [9], a modulation term is added to DMP to enable environment interaction. A force profile is encoded in attractor trajectory though admittance relationship in [10]. An explicit force profile is learned in the format of DMP separately from motion [11]. Robots gain the tactile skill to handle complex tasks by combining motion and force profiles.

A recent review on LfD [12] categorizes the demonstration method into three types: 1) *Kinesthetic teaching*, 2) *Teleoperation*, and 3) *Passive observation*. The passive observation method faces challenges when humans cannot access the robot because replicating the robot's environment is complicated. Kinesthetic teaching refers to a mode in which the robot is physically guided through a task by human operators through direct interaction. Although the kinesthetic teaching provides a user-friendly interface, distinguishing between human guiding and interaction forces with the environment involves complex algorithms. Consequently, force profiles must be taught separately through teleoperation in [11]. LfD via teleoperation enables the experts to transfer skills to a robot that is not easily accessible. By utilizing the tactile sensors of the operator's device [13], the external force in kinesthetic teaching is now divided into the operator-applied force and the environment's interaction force, simplifying the process of learning tactile skills.

A typical teleoperation system consists of a remote station where a robot executes the tasks and a human operator station with an input interface to control the remote robot. These input interfaces vary from a simple Mouse-Keyboard (MK) configuration [14] to intricate Avatar systems showcased

---

\*Equal contribution.

8th Conference on Robot Learning (CoRL 2024), Munich, Germany.

Table 1: Summary of related work

| Method | Motion learning | Force learning | Simultaneous learning | Online adaptation | Remote teaching |
|---|---|---|---|---|---|
| IPFS-KTHI[11] | ✓ | ✓ | ✗ | ✗ | ✗ |
| PAPM[25] | ✓ | ✗ | ✗ | ✓ | ✗ |
| FACT-DS[26] | ✗ | ✓ | ✗ | ✓ | ✗ |
| RASHIL-RTSM[27] | ✓ | ✗ | ✗ | ✗ | ✓ |
| **Ours** | ✓ | ✓ | ✓ | ✓ | ✓ |

in the ANA XPRIZE Competition [15]. A commonly used input interface is haptic devices, allowing experts to teach haptic guidance references [16] or contact-rich tasks [10, 17]. However, due to the morphological differences between the haptic device and the execution robot, demonstrations using the haptic device are not very intuitive [18].

In this paper, we design a teleoperation framework that consists of two identical robot arms. One is located in the workspace to execute the tasks after learning them; the other one interacts with the human operator to demonstrate the tactile skill and allow the human operator to feel the remote environment. Our framework with two identical robot arms merges the intuitiveness of kinesthetic teaching with the capability for force learning to address the problem mentioned earlier in teleoperation. The goal is to learn and adapt periodic tactile skills, encompassing motion and force profiles, from remote human demonstrations without interrupting skill execution. Compared to the traditional Leader-Follower teleoperation architecture, the leading role transits between the two robots according to the designed autonomy level allocation. During the teaching phase, the human side takes the lead, and once teaching is complete, the robot in the workspace resumes the leading role and executes the tactile skill autonomously. This role-switching mechanism is known as shared autonomy [19]. The autonomy level allocation relies on metrics such as human intention[20], human ability [21, 22], or confidence in the learned skills [23]. In our work, autonomy allocation depends on the user's intention and confidence in skill learning. Another benefit of autonomy allocation is that our framework allows humans to intervene and adjust the learned skill online. Typically, adapting to new skills occurs offline and needs an additional operation to reset the robot [24].

The contributions of this work are as follows:

- A tele-teaching framework that enables a manipulator to simultaneously learn force and motion profile, thereby acquiring tactile skill from a remote demonstration.

- A mechanism for online learning and adaptation of tactile skills without interruption through the proposed framework.

- The autonomy allocation based on operator intention and skill learning confidence provides intuitive interaction between operator and robot.

## 2   Related Work

The main differences between our work and other methods are listed in Table 1. The IPFS-KTHI method[11] teaches motion and force profiles separately. Motion profiles are taught through Kinesthetic teaching, and the force profile is taught through a haptic device, which could lead to a lack of synchronization between motion and force. The RASHIL-RTSM approach[27] integrates a human sensorimotor system into the robot control loop through a teleoperation setup for remote teaching. Inspired by this framework, we use a Teacher Robot instead of the human sensorimotor system for remote teaching. The method most closely related to our periodic motion learning module is PAPM[25], which focuses only on tactile skills that involve constant contact force rather than learning force profiles through demonstration. Additionally, we expand the autonomy allocation strategy from PAPM, initially designed for motion adaptation in a single robot manipulator, to facilitate simultaneous motion and force adaptation within our teleoperation framework. For learning about contact forces, FACT-DS [26] is the closest related work. It aims to develop a compensation model that corrects force errors caused by non-flat surfaces. Unlike FACT-DS, where force profiles are

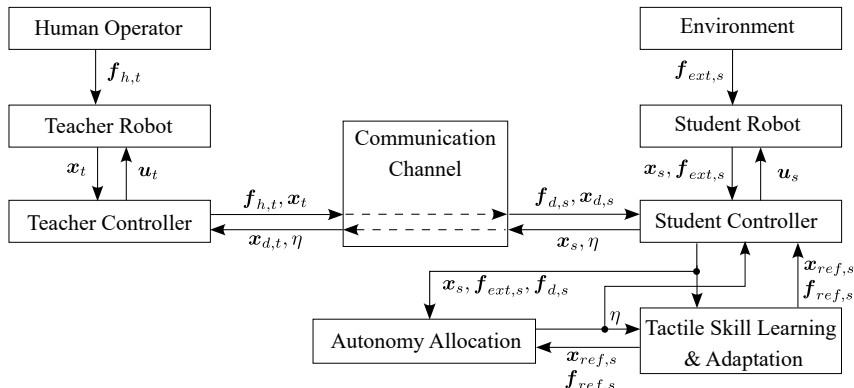

Figure 1: **Architecture block diagram for tele-teaching of tactile skill.**

position-dependent to adjust for these irregularities, in our work, force profiles are time-dependent, which allows for learning the forces shown by the user.

The primary innovation of our framework, distinguishing it from existing methods, is the simultaneous learning and adaptation of motion and force profiles—essential for tactile skills. This feature leverages the capabilities of our teleoperation framework, which can distinguish human guiding forces from external forces exerted on the robot without an additional force-torque sensor.

## 3 Methodology

### 3.1 Robot control in tele-teaching

The tele-teaching system consists of two identical robots to enhance the intuitiveness. The *Teacher Robot* (*TR*) is positioned at the human operator side, allowing humans to demonstrate tactile skills while feeling the tactile feedback of the remote environment. Conversely, the *Student Robot* (*SR*) is located remotely to execute the learned tactile skill. The control architecture described below illustrates how the Teacher Robot (*TR*) and Student Robot (*SR*) interact in terms of motion and forces to implement this tele-teaching system.

The sensory data of the robot systems including position $\boldsymbol{x}_t, \boldsymbol{x}_s$, measured external force $\boldsymbol{f}_{h,t}, \boldsymbol{f}_{ext,s}$, and the calculated autonomy level $\eta$ are exchanged through the communication channel and utilized within the controller (Figure 1). The dynamics of a gravity-compensated *TR* is written as,

$$\boldsymbol{M}_{c,t}(\boldsymbol{q}_t)\ddot{\boldsymbol{x}}_t + \boldsymbol{C}_{c,t}(\boldsymbol{q}_t, \dot{\boldsymbol{q}}_t)\dot{\boldsymbol{x}}_t = \eta\boldsymbol{u}_t + \boldsymbol{f}_{h,t}, \tag{1}$$

where $\boldsymbol{M}_{c,t} \in \mathbb{R}^{n \times n}$ and the $\boldsymbol{C}_{c,t} \in \mathbb{R}^{n \times n}$ are the inertia and Coriolis/centrifugal matrices of the *TR* represented in task space with $n$ dimensions, respectively. $\boldsymbol{x}_t \in \mathbb{R}^n$ is the *TR* end-effector position represented in the task space. $\boldsymbol{f}_{h,t} \in \mathbb{R}^n$ is the force applied by/to the human operator during the interaction with *TR* . $\boldsymbol{u}_t \in \mathbb{R}^n$ is the control command generated by the *TR* controller, and $\eta \in [0, 1]$ is the autonomy level that acts as an activation parameter to the controller. A $PD+$ controller is employed on *TR* to track the position of *SR* and provide the haptic feedback,

$$\boldsymbol{u}_t = \boldsymbol{K}_t\tilde{\boldsymbol{x}}_t + \boldsymbol{D}_t\dot{\tilde{\boldsymbol{x}}}_t, \tag{2}$$

where $\boldsymbol{K}_t \in \mathbb{R}^{n \times n}$ and $\boldsymbol{D}_t \in \mathbb{R}^{n \times n}$ represent the stiffness and the damping matrices, respectively. $\tilde{\boldsymbol{x}}_t = \boldsymbol{x}_{d,t} - \boldsymbol{x}_t$ denotes the trajectory following error between the *TR* and the desired trajectory $\boldsymbol{x}_{d,t} \in \mathbb{R}^n$, which is transmitted from *SR* .

On the *SR* side, similar dynamics with gravity compensation are written as,

$$\boldsymbol{M}_{c,s}(\boldsymbol{q}_s)\ddot{\boldsymbol{x}}_s + \boldsymbol{C}_{c,s}(\boldsymbol{q}_s, \dot{\boldsymbol{q}}_s) = \boldsymbol{u}_s + \boldsymbol{f}_{ext,s}, \tag{3}$$

where $\boldsymbol{M}_{c,s} \in \mathbb{R}^{n \times n}$ and the $\boldsymbol{C}_{c,s} \in \mathbb{R}^{n \times n}$ are the inertia and Coriolis/centrifugal matrices of the *SR* represented in task space, respectively. $\boldsymbol{u}_s \in \mathbb{R}^n$ denote the control command to *SR* :

$$\boldsymbol{u}_s = \eta\boldsymbol{u}_{ufic,s} + (1 - \eta)\boldsymbol{u}_{t,s} \tag{4}$$

where the $\boldsymbol{u}_{ufic,s}$ is the unified force/impedance control [28] to follow the reference tactile skill:

$$\boldsymbol{u}_{ufic,s} = \boldsymbol{u}_{m,s} + \boldsymbol{u}_{f,s}, \tag{5}$$

The control command $\boldsymbol{u}_{m,s}$ is the impedance controller for tracking motion profile,

$$\boldsymbol{u}_{m,s} = \boldsymbol{M}_{c,s}\ddot{\boldsymbol{x}}_{ref,s} + \boldsymbol{C}_{c,s}\dot{\boldsymbol{x}}_{ref,s} + \boldsymbol{K}_s\tilde{\boldsymbol{x}}_s + \boldsymbol{D}_s\dot{\tilde{\boldsymbol{x}}}_s \tag{6}$$

where $\tilde{\boldsymbol{x}}_s = \boldsymbol{x}_{ref,s} - \boldsymbol{x}_s$ is the tracking error and $\boldsymbol{K}_s \in \mathbb{R}^{n \times n}$ and $\boldsymbol{D}_s \in \mathbb{R}^{n \times n}$ are the impedance matrices. Moreover, $\boldsymbol{u}_{f,s}$ is a $PID$ controller for force tracking with feed-forward term,

$$\boldsymbol{u}_{f,s} = \boldsymbol{f}_{ref,s} + k_p\tilde{\boldsymbol{f}}_s + k_i \int \tilde{\boldsymbol{f}}_s dt + k_d\dot{\tilde{\boldsymbol{f}}}_s, \tag{7}$$

where $\tilde{\boldsymbol{f}}_s = \boldsymbol{f}_{ref,s} - \boldsymbol{f}_{ext,s}$ represents the tracking error for force. The passivity of the controller can be ensured by designing an energy tank[29], which provides an "energy budget" and releases energy for the execution of non-passive control actions by modifying the control command.

The control force $\boldsymbol{u}_{t,s}$ for the *SR* is designed to track the *TR* through both position and force channels, i.e.,

$$\boldsymbol{u}_{t,s} = \boldsymbol{K}_{t,s}(\boldsymbol{x}_{d,s} - \boldsymbol{x}_s) + \boldsymbol{D}_{t,s}(\dot{\boldsymbol{x}}_{d,s} - \dot{\boldsymbol{x}}_s) + \boldsymbol{f}_{d,s}, \cdot \tag{8}$$

Notably, for simplicity, only the translational dimensions are considered for the task space coordinate, i.e., $n = 3$. The orientation is constrained by high impedance, and the null space of the robot is controlled to maintain a specific joint configuration with best-effort in a compliant manner while ensuring task execution [30].

## 3.2   Tactile skill generation

The generated reference tactile skill comprises both motion and force profiles. For the reference motion profile, a periodic DMP formalism is adopted to generate the reference motion [7]:

$$\ddot{\boldsymbol{x}}_{ref,s} = \Omega^2(\alpha_x(\beta_x(\boldsymbol{x}_g - \boldsymbol{x}_{ref,s}) - \frac{\dot{\boldsymbol{x}}_{ref,s}}{\Omega}) + \boldsymbol{\gamma}_m(s)) \tag{9}$$

where $\alpha_x$ and $\beta_x$ are constants, along with the anchor point $\boldsymbol{x}_g \in \mathbb{R}^n$. Note that the force profile is not encoded by the DMP in Eq. 9 due to irreducible noise in the force reference signal $\boldsymbol{f}_{ext,s}$ from the sensor. Derivatives of the force signal, $\dot{\boldsymbol{f}}_{ext,s}$ and $\ddot{\boldsymbol{f}}_{ext,s}$, will be too noisy to be effectively encoded with DMPs. To ensure synchronization across different dimensions, the basic frequency of the generated tactile skill is determined as the minimum frequency value among all the dimensions.

$$\Omega = \min\{\omega_{1,m}, \omega_{2,m}, \ldots, \omega_{n,m}\} \tag{10}$$

The change rate of the canonical phase is the learned frequency, i.e., $\dot{s} = \Omega$. Additionally, $\boldsymbol{\gamma}_m$ is a linear combination of $N$ Radial Basis Functions (RBFs) as following,

$$\boldsymbol{\gamma}_m(s) = r\frac{\sum_{i=1}^{N} \mathbf{w}_{i,m}\psi_i(s)}{\sum_{i=1}^{N} \psi_i(s)}, \tag{11}$$

$$\psi_i(s) = \exp\left(h(\cos(s - c_i) - 1)\right), \ i = 1, 2, \cdots, N, \tag{12}$$

where, $c_i$ represents the center of RBF, $h$ denotes the width, and $r$ is the amplitude modulation parameter which is set to 1. Each vector $\mathbf{w}_{i,m} \in \mathbb{R}^n$ of the weights matrix $\mathbf{w}_m \in \mathbb{R}^{n \times N}$ is updated with the Recursive Least Square (RLS) algorithm [31]:

$$\mathbf{w}_{i,m}(t+1) = \mathbf{w}_{i,m}(t) + \boldsymbol{\psi}_i\mathrm{diag}(\boldsymbol{\sigma}_{i,m}(t+1))\boldsymbol{e}_{r,i,m}(t), \tag{13}$$

where $\boldsymbol{\sigma}_{i,m} \in \mathbb{R}^n$ is the vector of inverse covariance associated to the weights $\mathbf{w}_{i,m}$. Each element of $\boldsymbol{\sigma}_{i,m}$ is updated as

$$\boldsymbol{P}_{j,i,m}(t+1) = \frac{1}{\lambda_{fg}}\left(\boldsymbol{P}_{j,i,m}(t) - \frac{\boldsymbol{P}_{j,i,m}(t)^2}{\frac{\lambda_{fg}}{\boldsymbol{\psi}_{i,m}} + \boldsymbol{P}_{j,i,m}(t)}\right), \tag{14}$$

$\lambda_{fg}$ is the forgetting factor affecting the convergence speed. The initial value of $\boldsymbol{P}_{j,i,m}(0)$ is set to $1$ to avoid getting stuck at zero.

The goal of RLS is to minimize the error $\boldsymbol{e}_{r,i,m}$ which is defined as,

$$\boldsymbol{e}_{r,i,m}(t) = (1 - \eta)(\boldsymbol{\gamma}_{m,d}(t) - \mathbf{w}_{i,m}). \tag{15}$$

Notably, the term $(1 - \eta)$ ensures the error is set to zero when the *SR* executes autonomously, thus preventing updates of the weight matrix. The target is defined differently for motion and force. The motion profile is generated using DMP, and the target is defined as

$$\gamma_{d,m} = \frac{\ddot{\boldsymbol{x}}_t}{\Omega^2} - \alpha_x(\beta_x(\boldsymbol{x}_g - \boldsymbol{x}) - \frac{\dot{\boldsymbol{x}}_t}{\Omega}), \tag{16}$$

Note that the subscript $-m$ employed in the previous equations denotes their applicability to motion profile generation. For the force profile, due to the high noise of the first and second-order derivative of the measured force signal, the measured external force is directly assigned as the target value,

$$\gamma_{d,f} = \boldsymbol{f}_{ext,s}. \tag{17}$$

Substituting the subscript $-m$ with $-f$ in the above equations (excluding equation (9)), such that they are valid for force profile generation, the reference force profile is given as $\boldsymbol{f}_{ref,s} = \boldsymbol{\gamma}_f$.

## 3.3 Autonomy allocation

Autonomy allocation enables seamless transitions between human intervention and robot autonomy by assessing two factors: whether a skill has been successfully learned, and whether the user intends to end the current demonstration or start a new one. If the system determines that the skill has been learned and no new demonstrations are intended, it transitions to robot autonomy. Conversely, if the skill learning is incomplete or a new demonstration is indicated, the system reverts to human intervention mode. The autonomy level $\eta \in [0, 1]$ on the *SR* side indicates whether the human or the robot is leading during the learning phase. At $\eta = 0$, the *TR* complies with the human's motion, and the *SR* is compliant to follow the motion of *TR* . Throughout the learning and adaptation process, $\eta$ gradually smoothly changes the role of the robot system from compliantly following the human to following the generated reference tactile skill. After the skill learning and adaptation, the $\eta$ increases to 1, indicating the learning process ends and the robot system becomes autonomous until the next human intervention. This autonomy allocation also indicates the role changing between leader and follower in teleoperation system, as disscussed in Appendix 6.2.1

The rate of change of the autonomy level is defined as,

$$\dot{\eta} = \begin{cases} \max\{\eta_r, 0\}, & \eta = 0 \\ \eta_r, & 0 < \eta < 1 \\ \min\{\eta_r, 0\}, & \eta = 1, \end{cases} \tag{18}$$

with

$$\eta_r = \left(\frac{\eta}{\rho} + \epsilon\right)(1 - I_h - I_s), \tag{19}$$

where $I_h$ is a value that increases during human intervention for learning a new skill and given by,

$$I_h = \left(\frac{\|\boldsymbol{f}_{d,s}\|}{\lambda_1}\right)^3, \tag{20}$$

with appropriate constant $\lambda_1$. This way, by applying external force from the human demonstrator, the level of autonomy decreases to comply with the human. Moreover,

$$I_s = \left(\frac{(1 - \eta)\|\boldsymbol{e}_m\|}{\lambda_2}\right)^2 + \left(\frac{\|\tilde{\boldsymbol{x}}_s\|}{\lambda_3}\right)^4 + \left(\frac{\|\tilde{\boldsymbol{f}}_s\|}{\lambda_4}\right)^4, \tag{21}$$

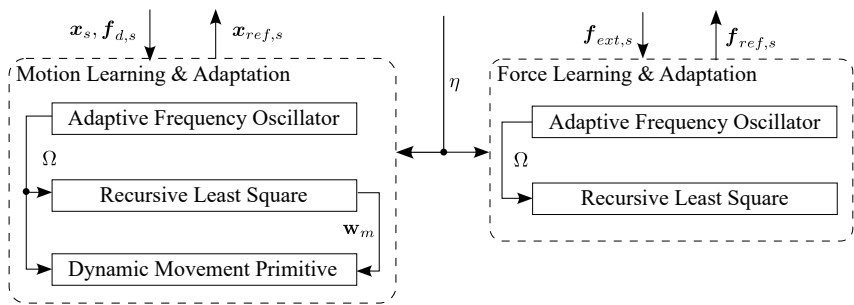

Figure 2: Block diagram of the *Tactile Skill Learning & Adaptation*. The motion and force profiles follow the same procedure. The force profile does not utilize the DMP due to the noise associated with its derivative.

which indicates the confidence of the *SR* in learning the tactile skill. Here, the $\| \, \boldsymbol{e}_m \, \|$ denotes the error vector, which will be introduced in the next section. During the learning phase, and as the motion tracking error $\tilde{\boldsymbol{x}}_s$ and force tracking error $\tilde{\boldsymbol{f}}_s$ gradually goes below a certain threshold $\lambda_3$ and $\lambda_4$, the value of $I_s$ decreases. This way, the confidence in learning increases, and thus, the autonomy level increases. To prevent getting stuck at zero, a small parameter $\epsilon$ is introduced in (19), while $\rho$ acts as a scaling parameter. A tuning guidance is provided in the Appendix 6.1.1.

### 3.4 Adaptive frequency oscillators

Adaptive frequency oscillators are tasked with identifying the basic frequency $\Omega$ of a demonstrated periodic skill in real time. During the learning and adaptation of a *periodic* tactile skill, the motion and the force undergo the same procedure, as illustrated in Figure 2. A generalized state $\boldsymbol{p}$ represents both motion and force state. In the following equations, the subscript $-_m$ stands for motion profile, and the state $\boldsymbol{p}_m = \boldsymbol{x}_s$. To adapt the frequency oscillators for the force profile, we need to replace the subscript with $-_f$ and go through the following formula again, and the state will be $\boldsymbol{p}_f = \boldsymbol{f}_{ext,s}$.

Following the same approach proposed in [32], the frequency oscillator is adapted by,

$$\dot{\boldsymbol{\theta}}_m = \boldsymbol{\omega}_m - (1 - \eta)\nu_m \boldsymbol{E}_m \sin(\boldsymbol{\theta}_m) \,, \tag{22}$$

$$\dot{\boldsymbol{\omega}}_m = -(1 - \eta)(\nu_m \boldsymbol{E}_m \sin(\boldsymbol{\theta}_m)) \,, \tag{23}$$

$$\boldsymbol{E}_m = \mathrm{diag}(\boldsymbol{p}_m - \hat{\boldsymbol{p}}_m) \,, \tag{24}$$

where $\boldsymbol{\theta}_m \in \mathbb{R}^n$ is the vector of the corresponding phase, $\boldsymbol{\omega}_m \in \mathbb{R}^n$ is the vector of frequencies, from which the basic frequency $\Omega$ is acquired as Equation (10) (the basic frequency $\Omega$ is the minimum), $\nu_m \in \mathbb{R}$ is the coupling strength and $\boldsymbol{E}_m \in \mathbb{R}^{n \times n}$ is a diagonal matrix considering the error between the input signal $\boldsymbol{p}_m$ and the estimation of the signal $\hat{\boldsymbol{p}}_m$. The initial values of $\boldsymbol{\omega}_m$ and $\boldsymbol{\theta}_m$ are set to a small value to avoid being stuck at zero.

Note that the bigger the coupling strength $\nu$, the faster the adaptive frequency oscillator converges [31], but the more unstable when $\omega$ is closer to zero. The learning rate $\mu_m$ also impacts the convergence speed and algorithmic stability. A detailed sensitivity analysis of coupling strength and learning rate and the tuning strategy is provided in Appendix 6.1.2.

The $i$th element of the estimation $\hat{\boldsymbol{p}}_m$ is computed as

$$\hat{p}_{i,m} = \sum_{k=0}^{M} (\alpha_{i,k,m} \cos(k\theta_{i,m}) + \beta_{i,k,m} \sin(k\theta_{i,m})), \ i = 1, 2, ..., n \,, \tag{25}$$

where $M$ denotes the number of Fourier components. The amplitudes $\alpha_{i,k,m}$, $\beta_{i,k,m}$ are updated as follows,

$$\begin{bmatrix} \dot{\alpha}_{i,k,m} \\ \dot{\beta}_{i,k,m} \end{bmatrix} = \begin{bmatrix} (1 - \eta) \, \mu_m \, \cos(k\theta_{i,m}) \, e_{i,m} \\ (1 - \eta) \, \mu_m \, \sin(k\theta_{i,m}) \, e_{i,m} \end{bmatrix} \,. \tag{26}$$

As the frequency $\Omega$ is the denominator in the DMP expression in Eq. 9 and 16, thus we set a lower bound to the frequency in the experiment for safety.

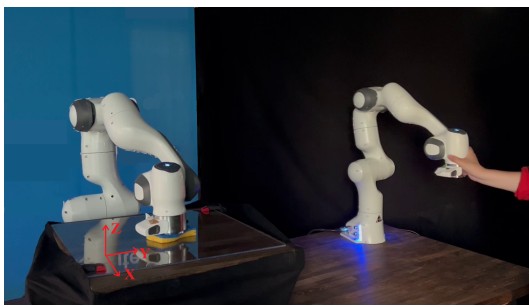

Figure 3: Experimental tele-teaching setup consists of two identical 7DOF robot arms. One robot executes a tactile skill, and the other robot is used to demonstrate or modify the skill over a teleoperation framework.

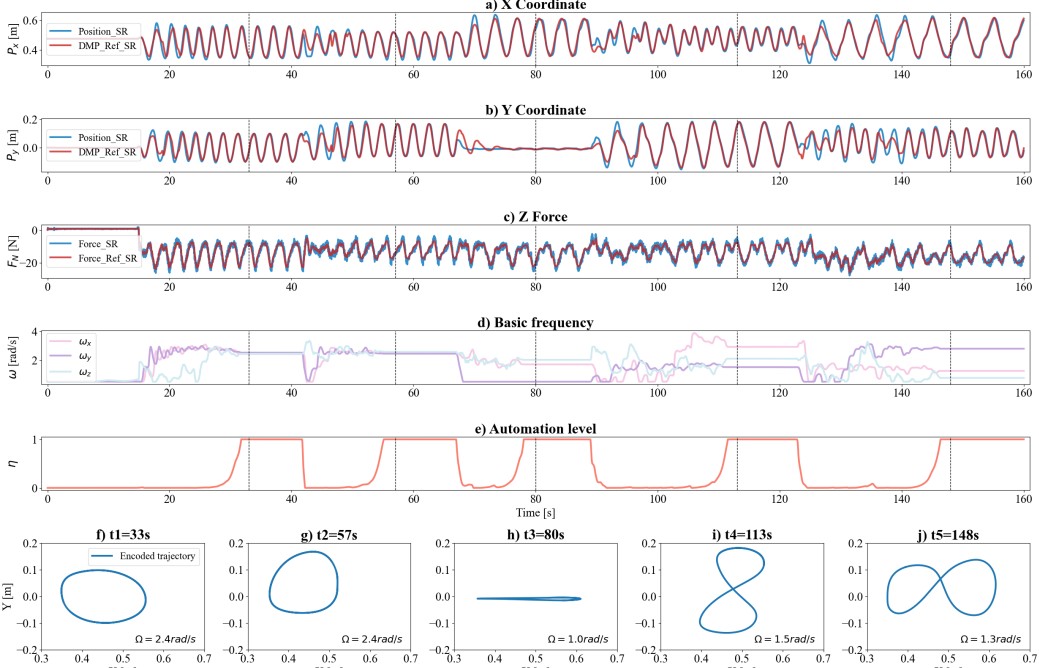

Figure 4: Experimental results of the proposed tele-teaching framework. A human operator demonstrates and modifies different periodic tactile skills to the *SR* through *TR* . a) and b) *SR*'s end-effector position $\boldsymbol{x}_s$ and DMP reference trajectory $\boldsymbol{x}_{ref,s}$, c) The contact force of *SR* $\boldsymbol{f}_{ext,s}$ and reference force $\boldsymbol{f}_{ref,s}$, d) Basic frequency of motion $\omega_x$, $\omega_y$ and of force $\omega_z$ and e) Autonomy level $\eta$. f) to j) The final encoded DMP trajectories on the XY plane for every learning cycle. The results show that the *SR* can quickly learn and adapt the tactile skill.

# 4 Experiments and Results

## 4.1 Experiment setup

The experimental setup consists of two 7 Degrees of Freedom (DoF) Franka Panda robots [33], as depicted in Figure 3. A sponge is fixed to the end-effector of the *SR* to interact with the table surface. The operator interacts with *TR* directly by grasping the robot end-effector. The data between *TR* and *SR* is transmitted through User Datagram Protocol (UDP) with negligible communication delay. The user demonstrates tactile skills on the *TR*, including motion on the $XY$ plane and force in the $Z$ direction. A series of different tactile skills is performed continuously to illustrate the simultaneous motion and force learning and adaptation capability of our tele-teaching framework. The values of the parameters used in the experiment can be found in Appendix 6.1.

## 4.2 Experiment procedure and results

The experiment encompasses the demonstration of five periodic tactile skills denoted as $S_1, \cdots, S_5$ with planar $XY$ motion and contact force in the $Z$ direction. Specifically, the skills contain,

- $S_1$: a clockwise circular motion with a large changing contact force,
- $S_2$: a counter-clockwise circular motion with a small changing contact force,
- $S_3$: a linear motion with forceful push and gentle pull,
- $S_4$: an $\infty$ shape along the $Y$ axis with reduced interaction force
- $S_5$: an $\infty$ shape along the $X$ axis with increased interaction force

The outcomes of the demonstrated tasks are depicted in Figure 4. We start the first demonstration of the $S_1$ at around $t = 16s$. After five iterations (around $t = 26s$) of repetition, the reference tactile skill converges to the user's demonstration, which suggests higher confidence in learning. Meanwhile, the basic frequency of the motion and force converge at the same value, around $\Omega = 2.4rad/s$. The convergence of the signal shape and frequency increases the autonomy level. The operator can feel that the robot becomes less compliant, indicating the acquisition of the new skill. After two more iterations at $t = 33s$, *SR* gains full autonomy and continuously reproduce the learned tactile skill. At around $t = 41s$, the operator initiates modifications to the previous skill, increasing the external force $\boldsymbol{f}_{h,t}$ on *TR*, and the tracking errors make $\eta$ drop quickly. Thus, it returns to the human demonstration mode and starts a new learning cycle. This process repeats for skills $S_2, \cdots, S_5$, respectively. [2]

Overall, it takes around six iterations ($16s$) to learn skill $S_1$ and $S_2$ with a minimal frequency at around $2.4rad/s$, three iterations ($6s$) for $S_3$ with $1.0rad/s$, and six iterations ($23s$) for $S_4$ and $S_5$ with $1.5rad/s$ and $1.3rad/s$. Notably, the frequency of $1.0rad/s$ in $S_3$ does not represent the actual frequency of the linear skill but rather the predefined minimal allowable frequency to prevent significant changes in DMP acceleration, as implicitly depicted in Eq. 16.

In summary, the proposed tele-teaching framework realizes online learning and adaptation of different tactile skills in only a few demonstration shots. The successful learning of the above five tactile skills, which differ spatially and temporally, verifies the proposed approach's effectiveness in simultaneous motion and force learning, online adaptation and remote teaching listed in Table 1. Furthermore, the autonomy allocation module obviates the need for robot resets between tactile skills, enabling seamless transitions between robot autonomy and human demonstration. This module empowers human operators to take over the control as needed, augmenting safety and reliability.

## 5    Conclusion and Future Work

This work develops a tele-teaching framework to enable real-time learning and adaptation of tactile skills from remote demonstration. Tactile skills demonstrated through the *Teacher Robot* are seamlessly transferred online to the *Student Robot* located remotely. Our approach distinctively integrates motion and force into a single cohesive tactile skill. The tele-teaching framework demonstrates remarkable speed and efficiency in adapting to new tactile skills. The adaptation occurs smoothly within just a few iterations. The autonomy allocation is based on the learning confidence and the operator's intentions. This strategy not only ensures smooth transitions between human demonstration and autonomous robot execution but also empowers the user to dictate the cessation or initiation of teaching phases.

We currently focus on simple periodic tactile skills with only translational motions and forces. Since the motion component is represented by Dynamic Movement Primitives (DMPs), the limitations associated with periodic DMPs are also applicable to our framework. Future works include exploring complex tactile skills that involve changes in orientation, potential stability issues led by communication delays, and expanding into deeper human-robot collaboration such as rehabilitation tasks.

---

[2]The experiment video is available at `https://youtu.be/LYnUJ0cYJgs`

**Acknowledgments**

We gratefully acknowledge the funding of the Lighthouse Initiative Geriatronics by LongLeif GaPa gGmbH (Project Y). The authors acknowledge the financial support by the Federal Ministry of Education and Research of Germany (BMBF) in the programme of "Souverän. Digital. Vernetzt." Joint project 6G-life, project identification number 16KISK002. This work was supported by the European Union's Horizon 2020 research and innovation programme as part of the project ReconCycle under grant no. 871352.

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

# 6 Appendix

## 6.1 Guidance for parameter tuning

In the table 2, we list all the parameters and their values and range used during the experiment. Note that we dynamically adjust the parameters within a specific range. In the table 3, we explain the effects of the key parameters for every module in our system, including autonomy allocation, adaptive frequency oscillators(AFO), and recursive least squares(RLS).

Table 2: Parameter values and ranges used during the experiment.

| Param | Value | Param | Value | Param | Value | Param | Value |
|---|---|---|---|---|---|---|---|
| $\lambda_1$ | 25 | $\lambda_2$ | 0.04 | $\lambda_3$ | 0.06 | $\lambda_4$ | 5 |
| $\nu_m$ | [10,100] | $\nu_f$ | 2 | $\mu_m$ | [0.02,0.8] | $\mu_f$ | [1,10] |
| $N$ | 50 | $M$ | 1 | $\lambda_{fg}$ | [0.99, 0.9996] | $\rho$ | 0.8 |
| $\alpha_x$ | 20 | $\beta_x$ | 5 | $\epsilon$ | 0.01 | $h$ | $\frac{N^2}{9}$ |

Table 3: Summary of the key parameters

| Module | Parameter | Feature |
|---|---|---|
| Autonomy Allocation | $\lambda_1$ | Sensitivity to new demo |
| | $\lambda_{2,3,4}$ | Tolerance for learning error |
| | $\rho$ | Smoothness of the transitions |
| Adaptive Frequency Oscillators | $\nu_{m,f}$ | Coupling strength: convergence speed and stability |
| | $\mu_{m,f}$ | Learning rate: convergence speed and stability |
| | $M$ | Number of Fouriers: complexity of the skill |
| Recursive Least Square | N | Number of RBFs: complexity of the skill |
| | $\lambda_{fg}$ | Forgetting factor: how sensitive to new coming data |

### 6.1.1 Autonomy allocation

The autonomy allocation module in our framework acts as a mechanism that is both aware of learning confidence and responsive to user intentions, employing heuristic methods. The assessment of whether a skill has been successfully learned depends on various factors, including learning errors related to motion, force, and frequency, as well as the external force applied to the Teaching Robot, which indicates user intentions, such as initiating a new demonstration. Parameters within the autonomy allocation module are configured to set thresholds based on these criteria.

Given that variations in sensor and robot hardware can affect performance, it is essential to adjust these thresholds to align with the system's specific characteristics. To assist in this calibration process, we provide guidance on effectively tuning the parameters involved in autonomy allocation.

$\lambda_1$ is a force threshold, affecting how sensitive to new demonstration intention. When the demonstration force on the Teacher Robot is larger than $\lambda_1$, the Teacher Robot will gain the leading role. indicating readiness to receive a new demonstration.

Other $\lambda_{2,3,4}$ indicates how tolerant to the learning error in frequency($\lambda_2$), motion($\lambda_3$), and force($\lambda_4$). As human demonstration always contains noises, large $\lambda_{2,3,4}$ means more tolerance/less sensitivity to learning error. Too small $\lambda_{2,3,4}$ can cause longer learning iterations, and too large $\lambda_{2,3,4}$ can cause insufficient learning of the tactile skills.

$\rho$ affects how smooth the transition between robot autonomy to human intervention mode. Larger $\rho$ leads to a more smooth transition. A common choice is set around 1.

### 6.1.2 Adaptive Frequency Oscillators

Coupling strength $\nu_{m,f}$ and learning rate $\mu_{m,f}$ are the critical parameters for adaptive frequency oscillators(AFO). They are related to the convergence speed, algorithm stability, and accuracy of the AFO. Regarding the number of fouriers $M$, we find that $M = 1$ is already enough for the periodic skills demonstrated in this paper.

A sensitivity analysis of these two parameters provides insights into how changes in coupling strength and learning rate impact the learning process of the motion's frequency. Note that the analysis also applies to learning the force's frequency. Figure. 5 illustrates the effect of different set-

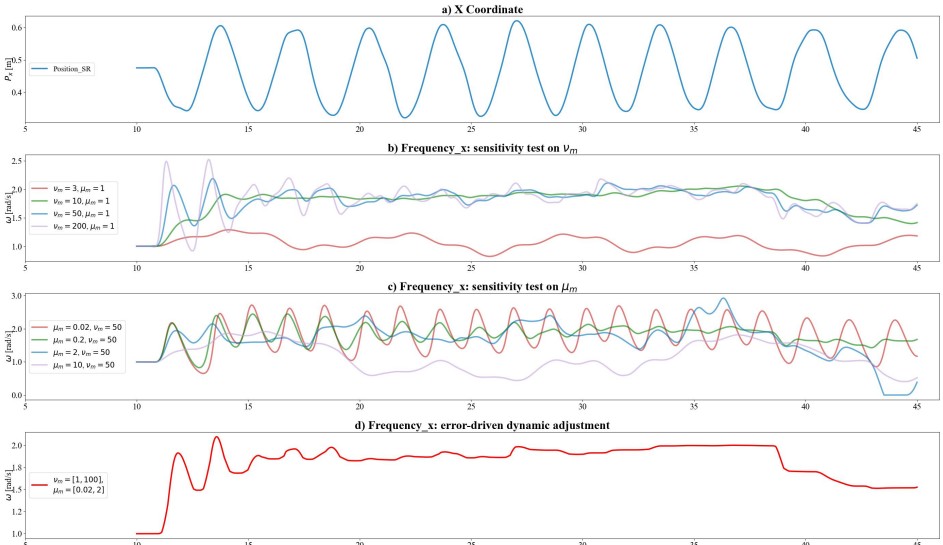

Figure 5: Comparison between various coupling strength $\nu_m$ and learning rate $\mu_m$ values within a single skill context. a) The target periodic demonstration. b) The estimated frequency of the target demonstration with various values of $\nu_m$ and a fixed $\mu_m$. c) The estimated frequency with various values of $\mu_m$ and a fixed $\nu_m$. d) The estimation of frequency through error-driven dynamic coupling strength and learning rate adjustments, where $\nu_m$ and $\mu_m$ gradually decrease from 100 and 2 to 1 and 0.02, respectively, as the error diminishes.

tings for coupling strength $\nu_m$ and learning rate $\mu_m$ on learning performance, with the target motion recorded from actual demonstrations of circular motions on the robot. The analysis of the coupling strength in the Figure. 5b) reveals that a higher coupling strength leads to quicker convergence but introduces larger initial oscillations. The analysis of the learning rate in the Figure. 5c) shows that a higher learning rate accelerates convergence but will cause instability when above a certain threshold. In summary, there is a trade-off between achieving rapid convergence and minimizing oscillations when tuning the coupling strength and learning rate.

Building on the insights from our sensitivity analysis, we propose an error-driven dynamic adjustment for the parameters $\nu_{m,f}$ and $\mu_{m,f}$. Note that here we use the error between input signal and the estimation of the signal in AFO. This strategy is designed to ensure rapid convergence when errors are large and to mitigate oscillations when errors are small. Specifically, both the coupling strength and learning rate are configured to decay progressively as the error decreases. The implementation of this dynamic adjustment is demonstrated Figure. 5d). The results indicate that, compared to scenarios with fixed parameters, this adaptive method maintains swift convergence while simultaneously minimizing oscillations, thus optimizing the learning process for adaptive frequency oscillators.

To show the sensitivity of coupling strength and learning rate in AFO when learning different skills. We test the performance of frequency learning with varying sets of parameters across the five periodic tactile skills demonstrated in the experiment section. As shown in the Figure. 6a), the frequency of motions in the y-dimension changes after every new demonstration. Figure 6d) reflects the robustness of our dynamic coupling strength and learning rate adjustment under different skills. Compared to the fixed setting shown in Figure 6b) and c), we observe faster convergence when adapting to a new skill and fewer oscillations(more accuracy) for the steady-state of learned frequency.

### 6.1.3  Recursive Least Square

The forgetting factor in the Recursive Least Squares (RLS) algorithm plays a crucial role in determining how much historical data influences current model updates. It dictates the size of a "sliding window" of historical data considered in the regression. For periodic skills, a good choice of the forgetting factor is intricately linked to the periods of the demonstrated skill. This relationship spec-

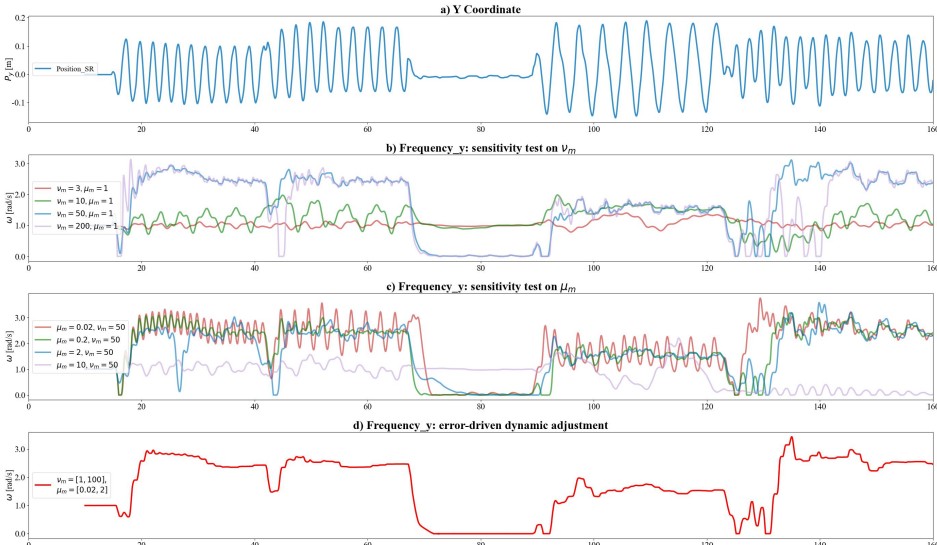

Figure 6: Comparison between various coupling strength $\nu_m$ and learning rate $\mu_m$ values across five different periodic skills. a) The target periodic demonstration. b) The estimated frequency with various values of $\nu_m$ and a fixed $\mu_m$. c) The estimated frequency with various values of $\mu_m$ and a fixed $\nu_m$. d) The estimation of frequency through error-driven dynamic coupling strength and learning rate adjustments, where $\nu_m$ and $\mu_m$ gradually decrease as the error diminishes.

ifies the number of demonstration cycles considered relevant for updating the learning model. An approximate relationship exists between the forgetting factor and the time window size over which historical data remains significant within the RLS framework. This relationship can be expressed mathematically, as shown in Equation 27. A larger forgetting factor expands the time window over which historical data is considered valid, thereby incorporating more past data in the regression.

$$\frac{1}{1 - \lambda_{fg}} = k \times \frac{1}{\Omega} \tag{27}$$

where $\frac{1}{1-\lambda_{fg}}$ is the estimated time window. $k$ decides how many rounds of demonstrations are considered. If $k$ is set to four, the learned skill is approximately an average of four last demonstrations.

The RLS adapts slowly to recent changes, with a high $\lambda_{fg}$. Conversely, a low $\lambda_{fg}$ makes RLS sensitive to new observations. To clearly show how the forgetting factor affects the learning procedure, we compare three sets of constant forgetting factors($k = 0.5, 2, 8$) under three nearly identical demonstrations of a circular movement on the x-y plane. It is challenging to maintain identical force profiles, so we omitted the force in this experiment. As shown in Figure 7a) and b), a small forgetting factor(when $k = 0.5$) helps quick convergence at the beginning of a new demonstration, trivial demonstration divergence will lead to shape distortion(between 21s and 27s). As the robot becomes less compliant with the user during the autonomy transition, it is hard for the user to demonstrate identical motions. Therefore, a larger forgetting factor is necessary during the transition to make it less sensitive to recent data. However, as shown in Figure 7e) and f), a larger forgetting factor(when $k = 8$) leads to less sensitivity to recent changes, resulting in a longer convergence time.

In conclusion, the optimal forgetting factor setting is a small value corresponding to $k = 0.5$ when the error is big, and a large value corresponds to, e.g., $k = 2 \sim 8$ when the error is small. The set of forgetting factors follows an error-driven dynamic adjustment, increasing for rapid convergence during significant errors and decreasing for stable autonomy transitions during minor errors. Note that here we use the error between the input signal and the learned signal in RLS. Regarding the number of RBFs $N$, we find that the common setting of $N$ ranging from 30 to 50 fits to the skills demonstrated in this paper.

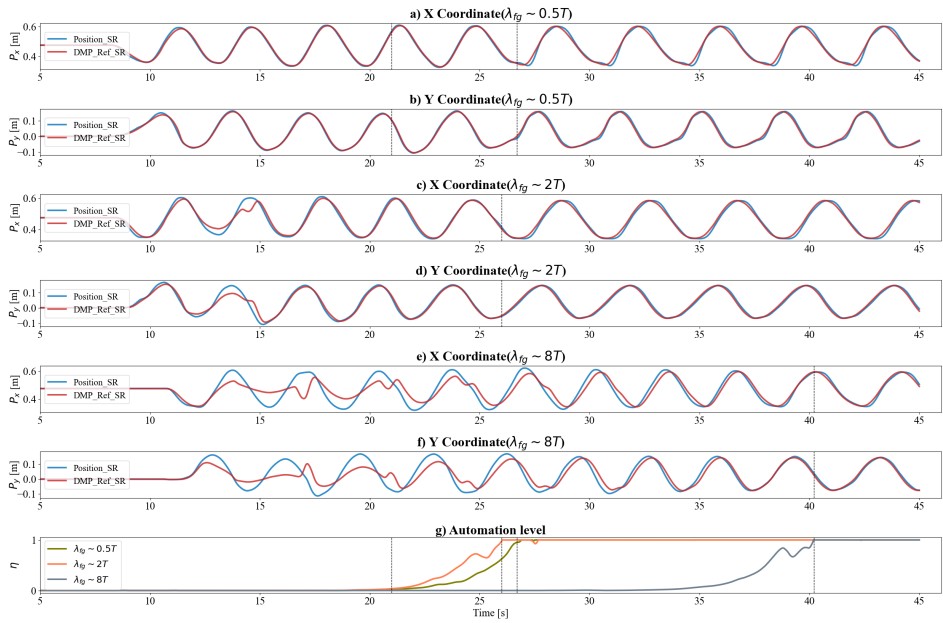

Figure 7: Comparison between different $\lambda_{fg}$.

## 6.2 Teleoperation system

### 6.2.1 Role allocation between *TR* and *SR*

The autonomy level $\eta$ dictates role allocation between the human and the robot system, resulting in the role change of the *TR* and *SR* . When $\eta = 0$, the *SR* completely follows the *TR* by both position and force channel. However, when the learning process finishes with $\eta = 1$, the *TR* transitions into a follower role, synchronizing its motion with the *SR* . Traditional leader-follower structure does not include the autonomy level, and *SR* only follows the *TR* by force. We realize that using both position and force channels from *TR* to *SR* is necessary in the tele-teaching framework. If only the force channel were used, minor position deviations would exist between *TR* and *SR* during the learning process($\eta$=0) because the two robot arms are not entirely identical in practice. As the autonomy level rises, the leading role transits from *TR* to *SR* . *TR* would follow *SR* gradually more in the motion channel, and the minor position deviations would be compensated. However, the human operator at *TR* 's side would feel that the *SR* learns a different motion. Therefore, the operator tends to naturally resist the compensation and keep the original demonstration, making the autonomy level $\eta$ fall again. The minor position deviations lead to unsmooth leading role transitions from *TR* to *SR* , resulting in unsuccessful tactile skill transfer. Furthermore, LfD via teleoperation presents certain drawbacks. Reference [18] shows that demonstrating a peg-in-hole task through teleoperation using a haptic device is more difficult for the operator than kinesthetic teaching. Users tend to demonstrate trajectory with higher variance. However, the method proposed in this work offers potential improvements. Another concern might be the stability issue with sub-optimal communication quality, leading to delays and packet loss. This problem can be alleviated by passivity-based methods like the Time Domain Passivity Approach (TDPA) [34, 35] or energy-based methods [36, 37].

