# OpenReview forum: "Online Transfer and Adaptation of Tactile Skill: A Teleoperation Framework"
_robot-learning.org/CoRL/2024/Conference — CoRL 2024_

### Official Review · Reviewer_encR · 2024-07-01
**Interesting contact-aware tele-operation system with a few important aspects missing**

**Originality:** 3
**Technical Quality:** 3
**Clarity Of Presentation:** 3
**Potential Impact:** 1
**Recommendation:** 3
**Confidence:** 4

**Review:**

This paper introduces an DMP-based framework for learning and adapting tactile-aware manipulation skills. The authors show that the system is able to automatically imitate a teacher robot, and it can then gradually switch from imitation to automation with some heuristics. The paper is overall well written. However, in my opinion a few important points are unclear:
Assumptions of the system are not clearly presented. E.g. the paper seems to assume that the learned skill is always periodical - what is the permissible range of period?
The paper lacks a sensitivity analysis or failure mode analysis. The system requires quite a few hyperparameters (16 according to Tab. 1), and it is unclear whether those parameters are robust and transferable between skills.
What is the underlying infrastructure for robot control?
Furthermore, in my opinion the paper’s impact is limited in that (1) it only applies to periodic skill learning and (2) the formulation of some important components of the system such as Eqn.20 and Eqn.21 is heavily based on heuristics. Although the paper presents an interesting framework that works well as it seems from the demo video, I do not think its topic aligns well with the main focus of CoRL.

In summary, I do not think this paper is suitable for publication at CoRL. Some more detailed comments:

Introduction: I’m not sure what “kinesthetic teaching” refers to and what is the difference between it and teleoperation. I assume it means scenarios where human operators directly move the robot arms - but it’s still quite ambiguous. Consider adding a short explanation.

Eqn. 2
TR’s control input u_t is calculated based on the positional difference between TR’s and SR’s trajectories. I’m wondering (1) instead of solely relying on delta positions, have the authors tried feeding back the force reading of SR to TR? (2) If positional difference is the only input for u_t, it is a traditional trajectory following system and in my opinion there is no haptic feedback from SR to TR and the text is a bit misleading.

“the null space of the robot is controlled according to the method outlined in [29]” - please provide an overview of the method as it is quite disruptive to scroll  to the end of the paper then scroll back.


Eqn.21
Consider using a different term for C_s, as C has already been used in Eqn.1 for Coriolis terms.

Eqn. 19
I’m also not sure why the confidence term is needed to determine the level of autonomy. If I understand it correctly, lower tracking errors leads to lower C_s, which leads to higher autonomy, which means the student robot relies more on replaying its learned skills rather than following the teacher’s motion - but why shouldn’t the student robot follows the teacher’s motion at all times as long as f_{d,s} is non-zero? Otherwise, wouldn’t the tracking error become larger and larger?

It is interesting to see the learning and adaptation processes of the 5 skills. However, I’m concerned about how sensitive the system is with regard to hyperparameters and whether a set of hyperparameters can adapt to a different skill. A sensitivity analysis, or a failure case analysis, which is actually required by CoRL (https://www.corl.org/contributions/call-for-papers), will be helpful.

**Quality Of The Limitations Section:**

1

**Questions For Rebuttal:**

Please see more details in the review above.

(1) What are the underlying assumptions of the proposed method?

(2) How sensitive are the hyperparameters when it comes to learning a different skill?

(3) What are the limitations of the learnable skills?

**Robotics Focus:**

4

**Summary Of Paper:**

This paper introduces an DMP-based framework for learning and adapting tactile-aware manipulation skills. The authors show that the system is able to automatically imitate a teacher robot, and it can then gradually switch from imitation to automation with some heuristics.

**Summary Of Recommendation:**

This paper presents an interesting control framework where a student robot imitates a teacher robot and can automatically switch to autonomy based on some heuristics. While the learned skills look interesting, I think a few very important aspects are missing, such as details on assumptions and a sensitivity analysis. Also, I do not think the paper’s topic aligns with the main focus of CoRL.

---

### Official Review · Reviewer_32qn · 2024-07-17
**Valuable contribution but the paper needs to be improved**

**Originality:** 3
**Technical Quality:** 4
**Clarity Of Presentation:** 2
**Potential Impact:** 3
**Recommendation:** 2
**Confidence:** 3

**Review:**

In general, the paper is well written even though I think the writing could still be improved.
The controller framework is nicely explained and contains sufficient detail, but the tactile skill generation section (2.2) and some other parts lack clarity. Here is the summary of my points:

Strengths:
- The control architecture in section 2.1, which is one of the core parts of the paper, is nicely explained
- According to the video, the approach seems to be working well and is a valuable contribution to the scientific community.

Major weaknesses:
- There is no distinct related work section. The introduction summarizes the state of the art but often the authors miss to highlight the differences between their work and the state of the art which makes it difficult to judge the novelty of the presented approach.
- The experiments do not contain any comparison with state of the art or a baseline method.
- I found the tactile skill generation section (2.1) a bit confusing. The authors present the DMP formulation and in the end of the section they mention that they do not really use DMPs for encoding the force profile. I suggest to mention in the beginning of the
section how motion and force are encoded before going into detail with the equations. The same is true for the beginning of Section 2. I think the reader would appreciate if the authors gave a short explanation of the individual parts (control architecture, frequency oscillator, DMPs, ..)
at the beginning of Section 2 before going into detail about the individual parts.
- The authors give an intuition about tuning the parameters in the discussion section but I wonder why no experiment was made about this so that the reader sees how learning performance is affected by some of the parameters.

Minor things:
- In the video there is a typo "dimenstions" instead of "dimensions"
- There are many equations in the paper which makes it difficult to maintain an overview over the type of variables (e.g., learned, parameter, input, output). Maybe a table with an overview would help here.

**Quality Of The Limitations Section:**

3

**Questions For Rebuttal:**

- Please correct "dimenstions" in video caption

- "Passivity can be ensured by energy tank": Could you add 1-2 sentences about this instead of just citing the paper?

- You mention that you do not encode rotation but your task-space is limited to the translational components? What modifications would be needed to go to 6D?

- It seems like the decision about which dimensions control force or motion is made beforehand and cannot be learned online. Is that true? What modifications would be necessary to support this?

- In line 166 you mention "without communication delay" which is of course not true. Maybe say "with minimal communication delay"?

- Why is there no comparison with a baseline method or with a state of the art method? If such a comparison is not feasible please elaborate on why you think that's the case.

**Robotics Focus:**

4

**Summary Of Paper:**

The authors present a framework for transferring skills from a human teacher to a robot. Hardware-wise, the setup consists of two panda arms. The human operator controls one of those arms while the student arm follows the arm reference force and motion profile generated by the operator. While doing this, a periodic DMP is trained in the background trying to mimic the motion/force profile. Over time, the DMP learns to mimic those profiles and the autonomy factor of the student arm increases, meaning the arm starts to perform the task on its own without human input. When the operator wants to change the motion, he/she can move the teacher arm again, leading to a decrease of the autonomy level of the student arm. Hence, the robot can interactively learn a new skill over time.

**Summary Of Recommendation:**

I think the approach is nice and according to the video and the evaluation it seems to work pretty well. I still think the paper could be made a lot stronger by adding some explanations and maybe improving the experiments section.

---

### Official Review · Reviewer_4fok · 2024-07-21
**Evaluating the Teleoperation Framework for Online Transfer and Adaptation of Tactile Skills**

**Originality:** 3
**Technical Quality:** 4
**Clarity Of Presentation:** 4
**Potential Impact:** 3
**Recommendation:** 3
**Confidence:** 4

**Review:**

### Quality
The quality of the paper is commendable, with a thorough explanation of the teleoperation framework and its components.

### Clarity
The paper is clear and concise, effectively conveying complex concepts in an accessible manner.

### Strengths:
1. The integration of DMP and RLS for tactile skill generation is a novel approach, providing a robust solution for tele-teaching.
2. The strategy based on learning confidence and operator intention ensures smooth transitions and effective human-robot collaboration.

### Weaknesses
1.  The paper briefly mentions future work on more complex tactile skills. The current experiments seem to focus on relatively simpler skills, and the framework's performance on more complex tasks is not demonstrated.
2.  While the framework shows promise in transitioning between human and robot control, deeper levels of human-robot collaboration are not explored in this paper. Further exploration could provide a more comprehensive understanding of the framework's capabilities.

**Quality Of The Limitations Section:**

2

**Questions For Rebuttal:**

1. Can the authors provide more details or preliminary results on how the framework performs with more complex tactile tasks? What are the anticipated challenges and how do they plan to address them?
2. The framework relies on remote operation. How does it handle potential issues arising from noise and latency in the communication channels? Are there any mitigations in place to address these challenges?
3. The success of DMPs and RLS often depends on the tuning of parameters. How sensitive is the framework to parameter settings, and what guidelines do the authors provide for optimal tuning?

**Robotics Focus:**

4

**Summary Of Paper:**

The paper presents an innovative teleoperation framework for the online learning and adaptation of tactile skills, eliminating the need for physical access to the execution robot. The proposed tele-teaching approach employs periodical Dynamical Movement Primitives (DMP) and Recursive Least Square (RLS) for generating tactile skills. The framework incorporates an autonomy allocation strategy guided by learning confidence and operator intention to ensure a smooth transition from human demonstration to autonomous robot operation. Experimental results using two 7 Degree of Freedom (DoF) Franka Panda robots demonstrate the framework's capability for online motion and force learning and adaptation within just a few iterations.

**Summary Of Recommendation:**

This paper presents a high-quality, innovative teleoperation framework for the online transfer and adaptation of tactile skills. The integration of Dynamical Movement Primitives (DMP) and Recursive Least Square (RLS) for tactile skill generation, coupled with a dynamic autonomy allocation strategy, demonstrates notable originality and significance. Empirical validation using advanced robotics hardware shows promising results in efficient skill adaptation. However, the framework's performance on more complex tasks, long-term stability, and deeper levels of human-robot collaboration need further exploration. Addressing these issues will further solidify the paper's contributions, making it a compelling addition to the field.

---

### Author Rebuttal · Authors · 2024-08-10

Dear Reviewers,


Thank you for your insightful feedback regarding our manuscript. This rebuttal is organized into two main sections to address the common concerns of reviewers and propose additional content to enhance the manuscript. The attached rebuttal letter contains a detailed comparison with related work and guidance for parameter tuning with sensitivity analysis.

In the **Comparison with Related Work** section, we highlight the major contributions of our study: the success of hardware demonstration addressing multiple features in online tactile skill learning. It also explains why direct comparisons with baseline methods were not feasible and details the distinct aspects of our approach. We compare our framework with the most closely related works to underscore the novelty of our work apart from the current state of the art.

The **Guidance for Parameter Tuning** section outlines the crucial parameters from different modules of our system. It starts with explaining the user-defined thresholds in the autonomy allocation module. We then discuss a sensitivity analysis of coupling strength and learning rate in adaptive frequency oscillators. We highlight the balance between convergence speed and algorithm stability and how these challenges are mitigated through error-driven dynamic adjustments. Additionally, we analyze the role of the forgetting factor in the recursive least square algorithm, presenting our methodology for its dynamic adjustment to optimize the learning process.

**Additional experiments for sensitivity analysis:**

+ Comparison between various coupling strength and learning rate values within a single periodic skill.
+ Comparison between various coupling strength and learning rate values across five periodic skills.
+ The effect of various forgetting values within a single periodic skill.

We would like to emphasize again that our approach has been successfully implemented in a robotic system, demonstrating the feasibility of online learning and adaptation to various tactile skills through remote demonstration. We believe this contribution will be of interest to the CoRL community.

---

We have updated the revised manuscript in the attachment.
Thanks again for your valuable reviews/comments!

Best regards,

The authors of this manuscript

---

### Decision · Program_Chairs · 2024-09-04

**Decision:**

Accept

**Comment:**

The authors propose a new method for teleoperation based on DMP. The method is new and the results are good. However, there are many confusing parts about the writing and we suggest the authors respond to the questions posted by the reviewers and improve the clarification of the paper.

-----
The authors addressed some questions posted by the reviewers. While the reviewers agree that the paper is of good quality and makes a good contribution, there is a concern about the weak correlation between the paper's method and CoRL.